# Generating Synthetic Datasets by Interpolating along Generalized Geodesics

**Jiaojiao Fan**[*]
Georgia Tech
jiaojiaofan@gatech.edu

**David Alvarez-Melis**
Microsoft Research
daalvare@microsoft.com

## Abstract

Data for pretraining machine learning models often consists of collections of heterogeneous datasets. Although training on their union is reasonable in agnostic settings, it might be suboptimal when the target domain —where the model will ultimately be used— is known in advance. In that case, one would ideally pretrain only on the dataset(s) most similar to the target one. Instead of limiting this choice to those datasets already present in the pretraining collection, here we explore extending this search to all datasets that can be synthesized as 'combinations' of them. We define such combinations as multi-dataset interpolations, formalized through the notion of generalized geodesics from optimal transport (OT) theory. We compute these geodesics using a recent notion of distance between labeled datasets, and derive alternative interpolation schemes based on it: using either barycentric projections or optimal transport maps, the latter computed using recent neural OT methods. These methods are scalable, efficient, and —notably— can be used to interpolate even between datasets with distinct and unrelated label sets. Through various experiments in transfer learning, we demonstrate this promising new approach to targeted on-demand dataset synthesis.

## 1 Introduction

Recent progress in machine learning has been characterized by the rapid adoption of large pretrained models as a fundamental building block (Brown et al., 2020). These models are typically pretrained on large amounts of general purpose data, and then adapted (e.g., fine-tuned) to a specific task of interest. Such pretraining datasets usually draw from multiple heterogeneous data sources, e.g., from arising from different domains or sources. Traditionally, all available datasets are used in their entirety during pretraining, for example by pooling them together into a single dataset (when they all share the same label sets) or by training in all of them sequentially one by one. These strategies, however, come with important disadvantages. Training on the union of multiple datasets might be prohibitive or too time consuming, and it might even be detrimental. Indeed, there is a growing line of research showing evidence that removing pretraining data sometimes helps transfer performance (Jain et al., 2022). On the other hand, sequential learning (e.g., consuming datasets one by one) is infamously prone to *catastrophic forgetting* (McCloskey & Cohen, 1989; Kirkpatrick et al., 2017): the information from earlier datasets gradually vanishing as the model is trained on new datasets. While all of these suggest that training on only some subset of the pretraining datasets, how to choose these is unclear. However, when the target dataset on which the model is to be used is known in advance, the answer is much easier: intuitively, one would train only on those relevant to the target one: e.g., those most similar to it. Indeed, recent work has shown that selecting pretraining datasets based on the distance to the target is a successful strategy (Alvarez-Melis & Fusi, 2020; Gao & Chaudhari, 2021). However, such methods are limited to selecting (only) among individual datasets already present in the collection.

---

[*]Work done while at Microsoft Research.

NeurIPS 2022 Workshop on Synthetic Data for Empowering ML Research.

In this work, we propose a novel approach to *generate* synthetic pretraining datasets as combinations of existing ones. In particular, this method searches among all possible continuous combinations of the available datasets, and thus is not limited to select one of them necessarily. When given access to the target dataset of interest, we seek among all such combinations the one closest (in terms of a metric between datasets) to the target. By characterizing datasets as sampled from a underlying probability distribution, this problem can be understood as a generalization (from Euclidean to probability space) of the problem of finding among the convex hull of a set of reference points, that closest to a query point. While this problem has a simple closed-form solution in Euclidean space (via an orthogonal projection), solving it probability space is, as we shall see here, much more challenging.

We tackle this problem from the perspective of interpolation. Formally, we model the combination of datasets as an interpolation between their distributions, formalized through the notion of geodesics in probability space endowed with the Wasserstein metric (Ambrosio et al., 2008; Villani, 2008). In particular, we rely on *generalized geodesics* (Craig, 2016; Ambrosio et al., 2008), constant-speed curves connecting a pair (or more) distributions parametrized with respect to a 'base' distribution, whose role is played by the target dataset in our setting. Computing such geodesics requires access to either an optimal transport coupling or map between the base distribution and every other reference distribution. The former can be computed very efficiently with off-the-shelf OT solvers, but are limited to generate only as many samples as the problem is originally solved on. In contrast, OT maps allow for on-demand out-of-sample mapping, and can be estimated using recent advances in neural OT methods (Fan et al., 2020; Korotin et al., 2022b; Makkuva et al., 2020). However, most existing OT methods assume unlabeled (feature-only) distributions, but our goal here is to interpolate between classification (i.e., labeled) datasets. Therefore, we leverage a recent generalization of OT to labeled datasets to compute couplings (Alvarez-Melis & Fusi, 2020), and adapt and generalize neural OT methods to the labeled setting to estimate OT maps.

In summary, the contributions of this paper are: (i) a novel approach to generate new synthetic classification datasets from existing ones by using geodesic interpolations, applicable even if they have disjoint label sets, (ii) two efficient methods to compute generalize geodesics, which might be of independent interest, (iii) empirical validation of the method in a transfer learning setting.

## 2 Background

### 2.1 Distributional interpolation with OT

Consider $\mathcal{P}(\mathcal{X})$ the space of probability distribution finite second moments over some Euclidean space $\mathcal{X}$. Given $\mu, \nu \in \mathcal{P}(\mathcal{X})$, the Monge formulation optimal transport problem seeks a map $T : \mathcal{X} \to \mathcal{X}$ that transforms $\mu$ into $\nu$ at minimal cost. Formally, the objective of this problem is $\min_{T:T\sharp\mu=\nu} \int_{\mathbb{R}^d} \|x - T(x)\|_2^2 \mathrm{d}\nu(x)$, where the minimization is over all the maps that pushforward distribution $\mu$ into distribution $\nu$. While a solution to this problem might not exist, a relaxation due to Kantorovich is guaranteed to have one. This modified version yields the 2-Wasserstein distance: $W_2^2(\mu, \nu) = \min_{\pi \in \Pi(\mu,\nu)} \int_{\mathbb{R}^d} \|x - x'\|_2^2 \mathrm{d}\pi(x, x')$, where now the constraint set $\Pi(\mu, \nu) = \{\pi \in \mathcal{P}(\mathcal{X}^2) \mid P_{0\sharp}\pi = \mu, P_{1\sharp}\pi = \nu\}$ contains all couplings with marginals $\mu$ and $\nu$. The optimal such coupling is known as the OT plan. A celebrated result by Brenier (1991) states that whenever $P$ has density with respect to Lebesgue measure, the optimal $T^*$ exists and is unique. In that case, the Kantorovich and Monge formulations coincide and their solutions are linked by $\pi^* = (\mathrm{Id}, T^*)_{\sharp}\mu$ where $\mathrm{Id}$ is the identity map. The Wasserstein-2 distance enjoys many desirable geometrical properties compared to other distances for distributions (Ambrosio et al., 2008). One such property is the characterization of geodesics in probability space (Agueh & Carlier, 2011; Santambrogio, 2015). When $\mathcal{P}(\mathcal{X})$ is equipped with metric $W_p$, the unique minimal geodesic between any two distributions $\mu_0$ and $\mu_1$ is fully determined by $\pi$, the optimal transport plan between them, through the relation:

$$\rho_t^D := ((1 - t)x + ty)\sharp\pi(x, y), \quad t \in [0, 1],$$

known as *displacement interpolation*. If the Monge map exists, the geodesic can also be written as

$$\rho_t^M := ((1 - t)\mathrm{Id} + tT^*)\sharp\mu_1, \quad t \in [0, 1], \tag{1}$$

and is known as *McCann's interpolation* (McCann, 1997). It easy to see that $\rho_0^M = \mu_1$ and $\rho_1^M = \mu_2$.

Such interpolations are only defined between two distributions. When there are $m \geq 2$ marginal distributions $\{\mu_1, \dots, \mu_m\}$, the *Wasserstein barycenter* $\rho_a^B := \arg\min_\rho \sum_{i=1}^m a_i W_2^2(\rho, \mu_i)$, $a \in \Delta_m \subset \mathbb{R}^m$ generalizes McCann's interpolation (Agueh & Carlier, 2011). Intuitively, the interpolation

parameters $a = [a_1, \ldots, a_m]$ determine the 'mixture proportions' of each dataset in the combination, akin to a convex combination of points in Euclidean space. In particular, when $a$ is a one-hot vector with $a_i = 1$, then $\rho_a^B = \mu_i$, i.e., the barycenter is simply the $i$-th distribution. Barycenters have attracted great attention in machine learning recently (Srivastava et al., 2018; Korotin et al., 2021), but they remain challenging to compute in high dimension (Fan et al., 2020; Korotin et al., 2022a).

Another limitation of these interpolation notions is the non-convexity of $W_2^2$ along them. In Euclidean space, given three points $x_1, x_2, y \in \mathbb{R}^d$, the function $t \mapsto \|x_t - y\|_2^2$, where $x_t$ is the interpolation $x_t = (1 - t)x_1 + tx_2$, is convex. In contrast, in Wasserstein space, neither the function $t \mapsto W_2^2(\rho_t^M, \nu)$ or $\mapsto W_2^2(\rho_a^B, \nu)$ are guaranteed to be convex (Santambrogio, 2017, Sec. 4.4). This complicates theoretical analysis, such as in gradient flows. To circumvent this issue, Ambrosio et al. (2008) introduced the *generalized geodesic* of $\{\mu_1, \ldots, \mu_m\}$ with base $\nu$ and defined it as $\rho_a^G := (\sum_{i=1}^m a_i T_i^*) \sharp \nu, \quad a \in \Delta_m$, where $T_i^*$ is the optimal map from $\nu$ to $\mu_i$.

**Lemma 1.** *The functional $\mu \mapsto W_2^2(\mu, \nu)$ is convex along the generalized geodesics, and* $W_2^2(\rho_a^G, \nu) \leq \sum_{i=1}^m a_i W_2^2(\mu_i, \nu)$.

Thus, unlike the barycenter the generalized geodesic does yield a notion of convexity satisfied by the Wasserstein distance, and is also easier to compute. For these reasons, we adopt this notion of interpolation for our approach. It remains to discuss how to apply it on (labeled) datasets.

## 2.2 Dataset distance

Consider a dataset $\mathcal{D}_P = \{z^{(i)}\}_{i=1}^N = \{x^{(i)}, y^{(i)}\}_{i=1}^N \overset{i.i.d.}{\sim} P(x, y)$. The Optimal Transport Dataset Distance (OTDD) (Alvarez-Melis & Fusi, 2020) measures its distance to another dataset $\mathcal{D}_Q$ as:

$$d_{\mathrm{OT}}^2(\mathcal{D}_P, \mathcal{D}_Q) = \min_{\pi \in \Pi(P,Q)} \int \left( \|x - x'\|_2^2 + W_2^2(\alpha_y, \alpha_{y'}) \right) \mathrm{d}\pi(z, z'), \tag{2}$$

which defines a proper metric between datasets. Here, $\alpha_y, \alpha_{y'}$ are class-conditional measures corresponding to $P(x|y)$ and $Q(x|y')$. This distance is strongly correlated with transfer learning performance, i.e., the accuracy achieved when training a model on $\mathcal{D}_P$ and then fine-tuning and evaluating on $\mathcal{D}_P$. Therefore, it can be used to select pretraining datasets for a given target domain. Henceforth we abuse the notation $P$ to represent both a dataset and its underlying distribution for simplicity. To avoid confusion, we use $\nu$ and $\mu$ to represent distributions in the feature space, which is Euclidean space, and use $P, Q$ to represent distributions in the product space of features and labels.

# 3 Method and algorithm

Our method consists of two steps: estimating optimal transport maps between the target dataset and all training datasets (Sec. 3.1), and using them to generate a convex combinations of these datasets by interpolating along generalized geodesics (Sec. 3.2). For some downstream applications we will additionally project the target dataset into the 'convex hull' of the training datasets (Sec. 3.3).

## 3.1 Solving optimal map between labelled datasets

The OTDD is a special case of Wasserstein distance, so it is natural to consider the alternative Monge (map-based) formulation to (2). We propose two methods to approximate the OTDD map, one using the entropy-regularized OT and another one based on neural OT.

**OTDD barycentric projection.** Barycentric projections (Ambrosio et al., 2008; Pooladian & Niles-Weed, 2021) can be efficiently computed for entropic regularized OT using the Sinkhorn algorithm (Sinkhorn, 1967). Assume that we have i.i.d. samples $X_\nu = (x_\nu^{(1)}, \ldots, x_\nu^{(N_\nu)}), X_\mu = (x_\mu^{(1)}, \ldots, x_\mu^{(N_\mu)})$ from two distributions $\nu$ and $\mu$ separately. After solving the optimal coupling $\pi^* := \min_{\pi \in \Pi(\nu,\mu)} \int \|x - x'\|_2^2 \mathrm{d}\pi(x, x')$, the barycentric projection can be expressed as $T_B(X_\nu) = N_\nu \pi^* X_\mu$. We extend the method to two datasets $Z_Q = \{X_Q, Y_Q\}, Z_P = \{X_P, Y_P\}$, where we have additional i.i.d. label data $Y_Q = (y_Q^{(1)}, \ldots, y_Q^{(N_Q)}), Y_P = (y_P^{(1)}, \ldots, y_P^{(N_P)})$. We first solve the optimal coupling $\pi^*$ for OTDD (2) following the regularized scheme in Alvarez-Melis & Fusi (2020), and represent labels as one-hot vectors $y \in \mathbb{R}^C$. The barycentric projection is divided into two parts:

$$\mathcal{T}_B(Z_Q) = [N_Q \pi^* X_P, N_Q \pi^* Y_P]. \tag{3}$$

However, this approach has two important limitations: it can not naturally map out-of-sample data and it does not scale well to large datasets (due to the quadratic dependency on sample size).

**OTDD neural map.** Inspired by recent approaches to estimate Monge maps using neural networks (Rout et al., 2022; Fan et al., 2021), we design a similar framework for the OTDD setting. Fan et al. (2021); Gazdieva et al. (2022) approach the Monge OT problem with general cost functions by solving its max-min dual problem $\sup_f \inf_T \int [c(x, T(x)) - f(T(x))] \, d\nu(x) + \int f(x') d\mu(x')$. We extend this method to the distributions involving labels by introducing an additional classifier in the map. Given two datasets $P, Q$, we parameterize the map $\mathcal{T}_N : \mathbb{R}^d \times \mathbb{R}^{C_Q} \to \mathbb{R}^d \times \mathbb{R}^{C_P}$ as

$$\mathcal{T}_N(z) = \mathcal{T}_N(x, y) = [\bar{x}; \bar{y}] = [G(z); \ell(G(z))],$$

where $G(\cdot) : \mathbb{R}^d \to \mathbb{R}^d$ is the pushforward feature map, and the $\ell(\cdot) : \mathbb{R}^d \to \mathbb{R}^{C_P}$ is a frozen classifier that is pre-trained on the dataset $P$. Notice that, with the cost $c(z, \mathcal{T}(z)) = \|x - G(z)\|_2^2 + W_2^2(\alpha_y, \alpha_{\bar{y}})$, the Monge formulation of OTDD (2) reads $\inf_{T\sharp Q = P} \int \|x - G(z)\|_2^2 + W_2^2(\alpha_y, \alpha_{\bar{y}}) dQ(z)$. We therefore propose to solve the max-min dual problem

$$\sup_f \inf_G \int \left[ \|x - G(z)\|_2^2 + W_2^2(\alpha_y, \alpha_{\bar{y}}) \right] dQ(z) - \int f(\bar{x}, \bar{y}) dQ(z) + \int f(x', y') dP(z'). \quad (4)$$

Implementation details are provided in Appendix D. Compared to previous conditional Monge map solvers (Bunne et al., 2022a; Asadulaev et al., 2022), the two methods proposed here: (i) do not assume class overlap across datasets, allowing for maps between datasets with different label sets; (ii) are invariant to class permutation and re-labeling; (iii) do not force no one-to-one class alignments, e.g., samples can be mapped across similar classes.

## 3.2 Convex combination in dataset space

Computing generalized geodesics requires constructing convex combinations of data points from different datasets. Given a weight vector $a \in \mathbb{R}^m$, features can be naturally combined as $x_a = \sum_{i=1}^m a_i x_i$. But combining labels is not as simple because: (i) we allow for datasets with different number of labels, so adding them directly is not possible; (ii) we do not assume different datasets have the same label sets, e.g. MNIST (digits) vs CIFAR10 (objects). Our solution is to represent all labels in the same dimensional space by padding them with zeros in all entries corresponding to other datasets. As an example, consider three datasets with $2, 3,$ and $4$ classes respectively. Given a label vector $y_1 \in \mathbb{R}^2$ for the first one, we embed it into $\mathbb{R}^9$ as $\tilde{y}_1 = [y_1; \mathbf{0}_3; \mathbf{0}_4]^\top$. Defining $\tilde{y}_2, \tilde{y}_3$ analogously, we compute their combination as $y_a = a_1 \tilde{y}_1 + a_2 \tilde{y}_2 + a_3 \tilde{y}_3$. This representation is loss-less and preserves the distinction of labels across datasets.

## 3.3 Projection onto generalized geodesic of datasets

We finally put together the components in Sec 3.1 and 3.2 to construct generalized geodesics between datasets in two steps. First, we compute OTDD maps $\mathcal{T}_i^*$ between $Q$ and all other datasets $P_i, i = 1, \ldots, m$ using the discrete or neural OT approaches. Then, for any interpolation vector $a \in \Delta_m$ we identify a dataset along the generalized geodesic via $P_a := (\sum_{i=1}^m a_i \mathcal{T}_i^*) \sharp Q$. By using the convex combination method in Sec. 3.2 for labeled data, we can efficiently sample from $P_a$.

We next consider locating the dataset $P_a^*$ that minimizes the distance between $P_a$ and $Q$, i.e. the projection of $Q$ onto the generalized geodesic. We firstly approach this problem from a Euclidean viewpoint. Suppose there are several distributions $\{\mu_i\}_{i=1}^m$ and an additional distribution $\nu$ on Euclidean space $\mathbb{R}^d$, Lemma 1 guarantees there exists a unique parameter $a^*$ that minimizes $W_2^2(\rho_a^G, \nu)$. However, it is not straightforward to locate $a^*$ because there is no closed-form formula of the map $a \mapsto W_2^2(\rho_a^G, \nu)$ and it can be expensive to calculate $W_2^2(\rho_a^G, \nu)$ for all possible $a$. To solve this problem, we resort to another transport distance: $(2,\nu)$-transport metric.

**Definition 1** (Craig (2016)). *The $(2,\nu)$-transport metric is given by* $W_{2,\nu}(\mu_i, \mu_j) := \left( \int \|T_i^*(x) - T_j^*(x)\|_2^2 d\nu(x) \right)^{1/2}$, *where $T_i^*$ is the optimal map from $\nu$ to $\mu_i$.*

When $\nu$ has density with respect to Lebesgue measure, then $W_{2,\nu}$ is a valid metric (Craig, 2016, Prop. 1.15). Moreover, we can derive the closed-form formula of the map $a \mapsto W_{2,\nu}^2(\rho_a^G, \nu)$.

**Proposition 1.** $W_{2,\nu}^2(\rho_a^G, \nu) = \sum_{i=1}^m a_i W_{2,\nu}^2(\mu_i, \nu) - \frac{1}{2} \sum_{i \neq j} a_i a_j W_{2,\nu}^2(\mu_i, \mu_j)$.

This equation implies that given distributions $\{\mu_i\}, \nu$ in Euclidean space, we can trivially solve the optimal $a^*$ that minimizes $W_{2,\nu}^2(\rho_a^G, \nu)$ by a quadratic programming solver[2]. The proof (Appendix C) relies on Brenier's theorem. Inspired by this, we also define a transport metric for datasets:

---

[2]We use the implementation https://github.com/stephane-caron/qpsolvers

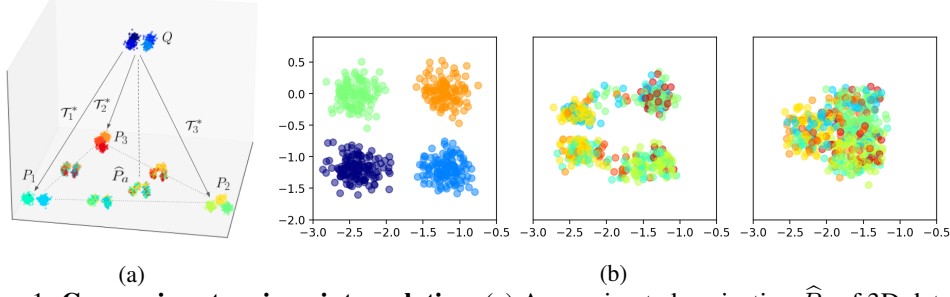

(a)                     (b)

Figure 1: **Comparison to mixup interpolation,** (a) Approximated projection $\widehat{P}_a$ of 3D dataset $Q$ onto the generalized geodesic of datasets $\{P_i\}$. (b) Left to right: the 2D projection of the datasets $Q$, $(\sum_{i=1}^{m} a_i^* \mathcal{T}_i^*) \sharp Q$, $(\sum_{i=1}^{m} a_i^* \mathcal{T}_i) \sharp Q$, where $\mathcal{T}_i^*$ is the (optimal) OTDD map and $\mathcal{T}_i$ uses mixup.

**Definition 2.** *The squared (2,Q)-dataset distance is given by* $\mathcal{W}_{2,Q}^2(P_i, P_j) :=$ $\int \left( \|x_i - x_j\|_2^2 + W_2^2(\alpha_{y_i}, \alpha_{y_j}) \right) dQ(z)$, *where* $[x_i; y_i] = \mathcal{T}_i^*(z)$ *and* $\mathcal{T}_i^*$ *is the OTDD map from* $Q$ *to* $P_i$.

Denote $\mathcal{P}_{2,Q}(\mathcal{X} \times \mathcal{P}(\mathcal{X}))$ as the set of all probability measures $P$ that satisfy $d_{\mathrm{OT}}(P, Q) < \infty$ and the OTDD map from $Q$ to $P$ exists. The following result shows that (2,Q)-dataset distance is a proper distance. The proof is also left to the Appendix C.

**Proposition 2.** $\mathcal{W}_{2,Q}$ *is a valid metric on* $\mathcal{P}_{2,Q}(\mathcal{X} \times \mathcal{P}(\mathcal{X}))$.

Unfortunately, in this case $\mathcal{W}_{2,Q}^2(P_i, P_j)$ does not have an analytic form like before because Brenier's theorem may not hold for a general transport cost problem. However, we still borrow this idea and define an approximated projection $\widehat{P}_a$ as the minimizer of function

$$\mathcal{W}^2(P_a, Q) := \sum_{i=1}^{m} a_i \mathcal{W}_{2,Q}^2(P_i, Q) - \frac{1}{2} \sum_{i \neq j} a_i a_j \mathcal{W}_{2,Q}^2(P_i, P_j), \tag{5}$$

which is an analog of Proposition 1. Unlike the Wasserstein distance, $\mathcal{W}_{2,Q}^2(\cdot, \cdot)$ is easier to compute because it does not involve optimization, so it is relatively cheap the locate the minimizer of $\mathcal{W}^2(P_a, Q)$. Experimentally, we observe that $W_{2,Q}^2(P_a, Q)$ is predictive of model transferability across tasks. Figure 1(a) illustrates this projection on toy 3D datasets, color-coded by class.

## 4 Experiments

### 4.1 Learning OTDD maps on synthetic datasets

Figure 1(b) illustrates the role of the optimal map in estimating the projection of a dataset into the generalized geodesic hull of three others. Using maps $\mathcal{T}_i^*$ estimated via barycentric projection (3) results in a better preservation of class structure, whereas using non-optimal maps $\mathcal{T}_i$ based on random couplings (as the usual *mixup* does) destroys class structure.

### 4.2 Transfer learning

Next, we use our framework to generate new pretraining datasets for few-shot learning. Given $m$ labeled pretraining datasets $\{P_i\}$, we consider a few-shot test dataset, in which only partial data is labelled, e.g. 5 samples per class. Suppose the training resource and time are both limited such that the user can choose only one dataset to train the model, in the mean time, the user expects the model to have the best generalization ability. To this end, we assume the training dataset is chosen from the generalized geodesic $\{P_a\}$. With a choice of the one-hot weight vector $a$, $P_a$ recovers the original dataset $P_i$ for some $i$. Otherwise, $P_a$ will be the interpolation of datasets $\{P_i\}$. We first show that the generalization ability of training models has a strong correlation with the distance $\mathcal{W}_{2,Q}^2(P_a, Q)$. Then we compare our framework with several baseline methods.

**Connection to generalization.** The closed-form expression of $W_{2,\nu}^2(\rho_a^G, \nu)$ (Prop. 1) provides the distance between a base distribution $\nu$ and the distribution along generalized geodesic $\rho_a^G$ in Euclidean space. We study its analog (5) for labelled datasets $Q$ and $\{P_i\}$ in Figure 2. To investigate

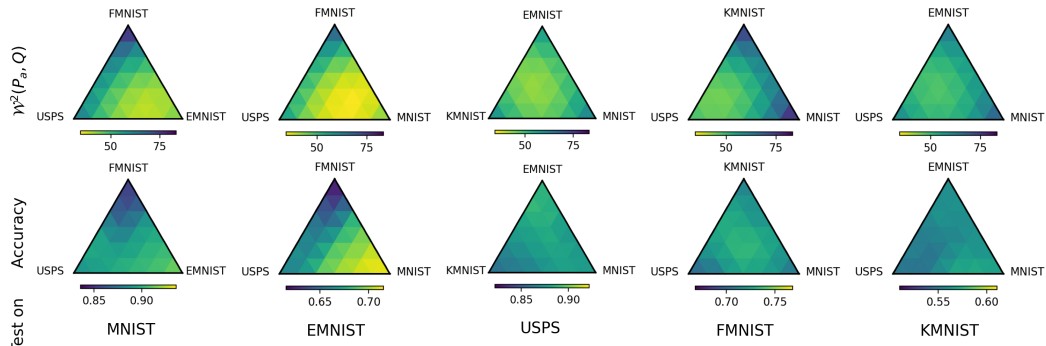

Figure 2: Relationship between the function $\mathcal{W}^2(P_a, Q)$ and the accuracy of the fine-tuned model. The training datasets are marked on the vertexes of each ternary plot. Different location in each ternary plot represents different interpolation dataset $P_a$, where the center is the most mixed dataset, i.e. $a = [1/3, 1/3, 1/3]$. We visualize the function (5) in the first row and the fine-tuning accuracy in the second. Comparing the first row and the second, we find the accuracy and $\mathcal{W}^2(P_a, Q)$ are highly correlated. This implies that the model trained on the minimizer dataset of $\mathcal{W}^2(P_a, Q)$ tends to have a better generalization ability. Each ternary plot is an average of 5 runs with distinct random seeds.

Table 1: **Pretraining on synthetic data**. Shown is 5-shot transfer accuracy (mean $\pm$ s.d. over 5 runs).

| Methods | MNIST-M | MNIST | USPS | FMNIST | KMNIST | EMNIST |
|---|---|---|---|---|---|---|
| OTDD barycentric projection | **42.10±4.37** | **93.74±1.46** | 86.01±1.50 | **70.12±3.02** | **52.55±2.73** | **67.06±2.55** |
| OTDD neural map | 40.06±4.75 | 88.78±3.85 | 83.80±1.60 | 70.02±2.59 | 50.32±3.10 | 65.32±1.80 |
| Mixup | 33.85±2.22 | 88.68±1.57 | **88.61±2.00** | 66.74±3.79 | 48.16±3.38 | 60.95±1.38 |
| Train on few-shot dataset | 19.10±3.57 | 72.80±3.10 | 80.73±2.07 | 60.50±3.07 | 41.67±2.11 | 53.60±1.18 |
| 1-NN on few-shot dataset | 20.95±1.39 | 64.50±3.32 | 73.64±2.35 | 60.92±2.42 | 40.18±3.09 | 39.70±0.57 |

the generalization abilities of models trained on different datasets, we discretize the simplex $\Delta_3$ to obtain 36 interpolation parameters $a$, and train a 5-layer LeNet classifier on each $P_a$. Then we fine-tune all of these classifiers on the few-shot test dataset $Q$ with only 20 samples per each class. We control the same number of training iterations and fine-tuning iterations across all experiments. We fix the same colorbar range for all heatmaps across datasets to highlight the different impact of choosing training dataset. For some test datasets, the choice of training dataset can affect the fine-tuning accuracy greatly. For example, when $Q$ is EMNIST and the training dataset is FMNIST, the fine-tuning accuracy is only $\sim 60\%$, but this can be improved to $> 70\%$ by choosing an interpolated dataset closer to MNIST. This is reasonable because MNIST shares more similarity with EMNSIT than FMNIST or USPS. To some test datasets like FMNIST and KMNIST, this difference is not so obvious because all training datasets are all far away from the test dataset.

**Comparison with baselines.** Next, we compare our method with several baseline methods on NIST datasets. In each set of experiment, we select one dataset as the test dataset, and the rest NIST datasets are the training datasets. We assume the test dataset is 5-shot, and to do this, we randomly choose 5 samples per class to be the labeled data, and treat the remaining samples as unlabeled. Our method firstly trains a model on $\widehat{P}_a$, and fine-tune the model on 5-shot test dataset. To obtain $\widehat{P}_a$, we use barycentric projection or neural map to approximate the OTDD maps from test dataset to the training datasets. Our results are shown in the first two rows in Table 1. The first baseline method is to create a synthetic dataset as training dataset by Mixup among datasets. We randomly sample data from each training datasets, and do the convex combination of them with weight $\hat{a}$. We use the same convex combination method in Sec. 3.2, thus this baseline is equivalent to our framework with suboptimal OTDD maps. The other two baselines (the bottom block in Table 1) skip the transfer learning part, and directly train the model or solve 1-NN on the few-shot test dataset. Overall, transfer learning can bring additional knowledge from other domains and improve the test accuracy by at most 21%. Among the methods in the first block, training on datasets generated by OTDD barycentric projection outperforms others except USPS dataset, where the difference is only about 2.6%.

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

## A   Related work

**Mixup and related In-Domain Interpolation**   Generating training data through convex combinations was popularized by *mixup* (Zhang et al., 2018): a simple data augmentation technique that interpolates features and labels between pairs of points. This and other works based on it (Zhang et al., 2021; Chuang & Mroueh, 2021) use mixup to improve in-domain model robustness and generalization by increasing in-distribution diversity of the training data. Although sharing some intuitive principles with mixup, our method interpolates entire datasets —rather than individual datapoints— with the goal of improving across-distribution diversity and therefore out-of-domain generalization.

**Dataset synthesis in machine learning**   Generating data beyond what is provided as training dataset is a crucial component of machine learning in practice. Basic transformations such as rotations, cropping, and pixel transformations can be found in most state-of-the-art computer vision models. More recently, Generative Adversarial Nets (GAN) have been used to generate synthetic data in various contexts (Bowles et al., 2018; Yoon et al., 2019), a technique that has proven particularly successful in the medical imaging domain (Sandfort et al., 2019). Since GANs are trained to replicate the dataset on which they are trained, these approaches are typically confined to generate in-distribution diversity, and typically operate on features only.

**Discrete OT, Neural OT, Gradient Flows**   Barycentric projection (Ambrosio et al., 2008; Perrot et al., 2016) is a typical effective method to approximate optimal transport map with discrete regularized OT. Other than this, neural net based optimal map in Euclidean space has made great progress (Makkuva et al., 2020; Fan et al., 2021; Rout et al., 2022) recently and reveal its power in image generation (Rout et al., 2022), style transfer (Korotin et al., 2022b), etc. However, the study of the optimal map between two datasets is relatively scarse. Some conditional Monge map solvers (Asadulaev et al., 2022; Bunne et al., 2022a) utilize the label information in a semi-supervised manner, where they assume the label to label correspondence between two distributions is known. Our dataset mapping is distinct from them because we do not enforce the label to label mapping. Based on the optimal coupling or map, geodesics and interpolation in general metric spaces have been studied extensively in the optimal transport and metric geometry literatures (McCann, 1997; Agueh & Carlier, 2011; Ambrosio et al., 2008; Santambrogio, 2015; Villani, 2008; Craig, 2016), albeit mostly in a theoretical setting. Gradient flows (Santambrogio, 2015), as an alternative approach for interpolation between distributions, have become increasingly popular in machine learning to model existing processes (Bunne et al., 2022b; Mokrov et al., 2021; Fan et al., 2022) or solving optimization problems over datasets (Alvarez-Melis & Fusi, 2021), but they are computationally heavy.

## B   Discussion

**Complexity**   The complexity of solving OTDD barycentric projection by Sinkhorn algorithm is $\mathcal{O}(N^2)$ (Dvurechensky et al., 2018), where $N$ is the number of data in both datasets. This can be expensive for large-scale dataset. In practice, we solve the batched barycentric projection, i.e. take a batch from source and target datasets and solve the projection from source batch to target batch, and we normally fix batch size $B$ as $10^4$. This reduces the complexity from $\mathcal{O}(N^2)$ to $\mathcal{O}(BN)$. The complexity of solving OTDD neural map is $\mathcal{O}(BKH)$, where $K$ is number of iterations, and $H$ is the size of the network. We always choose $K = \mathcal{O}(N)$ in the experiments. The complexity of solving all the $(2, Q)$-dataset distances in (5) is $\mathcal{O}(m^2 N)$ since we need to solve the dataset distance between each pair of training datasets. Putting these pieces together, the complexity of approximating the interpolation parameter $\hat{a}$ for the minimizer of (5) is $\mathcal{O}(N(B + m^2))$.

**Limitation**   The generation of synthetic dataset relies on solving OTDD maps from test dataset to each training dataset. These OTDD maps are tailored to the considered test dataset and can not be reused for a new test dataset. Another limitation is our framework is based on model training and fine-tuning pipeline. This can be resource demanding for large-scale models, like GPT model.

## C Proofs

*Proof of Lemma 1.* By Santambrogio (2017, Sec. 4.4), the result holds when $m = 2$. Then Proposition 7.5 in Agueh & Carlier (2011) extends the result to the case of $m > 2$. $\qquad\square$

*Proof of Proposition 1.* Since linear combination preserves cyclically monotonicity, $\sum_{i=1}^{m} a_i T_i^*(x)$ is the optimal map from $\nu$ to $\rho_a^G$ (McCann, 1995). Then according to the definition of $W_{2,\nu}(\cdot,\cdot)$, we can write

$$W_{2,\nu}^2(\rho_a^G, \nu) = \int \left\| x - \sum_{i=1}^{m} a_i T_i^*(x) \right\|^2 \mathrm{d}\nu(x). \tag{6}$$

For scalars $p, q_1, \ldots, q_m$, it holds that

$$\left( p - \sum_{i=1}^{m} a_i q_i \right)^2 = p^2 + \sum_{i=1}^{m} a_i^2 q_i^2 - 2\sum_{i=1}^{m} a_i p q_i + \sum_{i \neq j} a_i a_j q_i q_j$$

$$= p^2 + \sum_{i=1}^{m}(a_i - a_i \sum_{j \neq i} a_j) q_i^2 - 2\sum_{i=1}^{m} a_i p q_i + \sum_{i \neq j} a_i a_j q_i q_j$$

$$= \sum_{i=1}^{m} a_i(p - q_i)^2 - \frac{1}{2}\sum_{i \neq j} a_i a_j (q_i - q_j)^2.$$

Plugging this equality into (6) gives

$$W_{2,\nu}^2(\rho_a^G, \nu) = \int \left( \sum_{i=1}^{m} a_i \|x - T_i^*(x)\|^2 - \frac{1}{2}\sum_{i \neq j} a_i a_j \|T_i^*(x) - T_j^*(x)\|^2 \right) \mathrm{d}\nu(x)$$

$$= \sum_{i=1}^{m} a_i \int \|x - T_i^*(x)\|^2 \mathrm{d}\nu(x) - \frac{1}{2}\sum_{i \neq j} a_i a_j \int \|T_i^*(x) - T_j^*(x)\|^2 \mathrm{d}\nu(x)$$

$$= \sum_{i=1}^{m} a_i W_{2,\nu}^2(\mu_i, \nu) - \frac{1}{2}\sum_{i \neq j} a_i a_j W_{2,\nu}^2(\mu_i, \mu_j).$$

$\qquad\square$

*Proof of Proposition 2.* Firstly, $\mathcal{W}_{2,Q}$ is symmetric and nonnegative by definition. It is non-degenerate since $\mathcal{W}_{2,Q}(P_i, P_j) \geq d_{\mathrm{OT}}(P_i, P_j)$ and $d_{\mathrm{OT}}$ is a metric. Finally, we show it satisfies the triangular inequality. Indeed,

$$\mathcal{W}_{2,Q}(P_1, P_3)$$

$$= \left( \int \|x_1 - x_3\|^2 + W_2^2(\alpha_{y_1}, \alpha_{y_3}) \mathrm{d}Q(z) \right)^{1/2}$$

$$\leq \left( \int (\|x_1 - x_2\| + \|x_2 - x_3\|)^2 + (W_2(\alpha_{y_1}, \alpha_{y_2}) + W_2(\alpha_{y_2}, \alpha_{y_3}))^2 \mathrm{d}Q(z) \right)^{1/2}$$

$$\leq \left( \int \|x_1 - x_2\|^2 + W_2^2(\alpha_{y_1}, \alpha_{y_2}) \mathrm{d}Q(z) \right)^{1/2} + \left( \int \|x_2 - x_3\|^2 + W_2^2(\alpha_{y_2}, \alpha_{y_3}) \mathrm{d}Q(z) \right)^{1/2}$$

$$= \mathcal{W}_{2,Q}(P_1, P_2) + \mathcal{W}_{2,Q}(P_2, P_3),$$

where the first inequality is the triangular inequality and the second inequality is the Minkowski inequality. $\qquad\square$

## D Implementation details of OTDD map

**OTDD barycentric projection** We use the implementation https://github.com/microsoft/otdd to solve OTDD coupling. The rest part is straightforward.

**OTDD neural map**    To solve the problem (4), we parameterize $f, G, \ell$ to be three neural networks. In NIST dataset experiments, we parameterize $f$ as ResNet [3] from WGAN-QC (Liu et al., 2019), and take feature map $G$ to be UNet[4] (Ronneberger et al., 2015). We generate the labels $\bar{y}$ with a pre-trained classifier $\ell(\cdot)$, and use a LeNet or VGG-5 with Spinal layers[5] (Kabir et al., 2022) to parameterize $\ell(\cdot)$. In 2D Gaussian mixture experiments, we use Residual MLP to represent all of them.

We remove the discriminator's condition on label to simplify the loss function as

$$\sup_{f} \inf_{G} \int \Big( \underbrace{\|x - G(z)\|_2^2}_{\text{feature loss}} + \underbrace{W_2^2(\alpha_y, \alpha_{\bar{y}})}_{\text{label loss}} \Big) \mathrm{d}Q(z) \underbrace{- \int f(\bar{x}) \mathrm{d}Q(z) + \int f(x') \mathrm{d}P(z')}_{\text{discriminator loss}}.$$

In this formula, we assume both $y$ and $\bar{y}$ are hard labels, but in practice, the output of $\ell(\cdot)$ is a soft label. Simply taking the `argmax` to get a hard label can break the computational graph, so we replace the label loss $W_2^2(\alpha_y, \alpha_{\bar{y}})$ by $y^\top M \bar{y}$, where $M \in \mathbb{R}^{C_Q \times C_P}$ is the label-to-label matrix where $M(i,j) := W_2^2(\alpha_{y_i}, \alpha_{y_j})$, and $y$ is the one-hot label from dataset $Q$. The matrix $M$ is precomputed before the training, and is frozen during the training.

We pre-train the feature map $G$ to be identity map before the main adversarial training. We use the Exponential Moving Average[6] of the trained feature maps as the final feature map.

**Data processing**    For all the NIST datasets, we rescale the images to size $32 \times 32$, and repeat their channel 3 times and obtain 3-channel images. We use the default train-test split from `torchvision`.

**Hyperparameters**    For the experimental results in Sec. 4.2, we use the OTDD neural map and train them with learning rate $10^{-3}$ and batch size 64. We train a LeNet for 2000 iterations, and fine-tune for 100 epochs. Regard the comparison with other baselines in Sec. 4.2, for transfer learning methods, we train a SpinalNet for $10^4$ iterations, and fine-tune it for 2000 iterations on test dataset. Training from scratch on the test dataset takes also 2000 iterations.

# E    Additional results

## E.1    OTDD neural map visualization

In Figure 4, we in addition provide qualitative results of OTDD map from EMNIST (letter) (Cohen et al., 2017) dataset to all other *NIST dataset and USPS dataset. At this point, we can confirm three traits of OTDD map, which are mentioned at the end of Sec. 3.1.
1) We don't assume a known source label to target label correspondence. So we can map between two irrelevent datasets such as EMNIST and FashinMNIST. 2) The map is invariant to the permutation of label assignment. For example, we show two different labelling in Figure 3, and the final OTDD map will be the same. 3) It doesn't enforce the label to label mapping but would follow the feature similarity. From Figure 4 in the appendix, we notice many cross-class mapping behaviors. For example, when the target domain is USPS (Hull, 1994) dataset, the lower-case letter "l" is always mapped to digit 1, and the capital letter "L" is mapped to other digits such as 6 or 0 because the map follows the feature similarity.

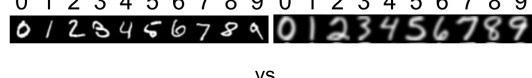

Figure 3: The numbers above images are the labels. In the first labelling method, all 0 MNIST digits are assigned as class "0", and they are labelled as class "7" in the bottom labelling.

We also show the OTDD neural map between 2D Gaussian mixture models with 16 components in Figure 5. This example is very special so that we have the closed-form solution of OTDD map. The feature map is a identity map and the pushforward label is equal to the corresponding class that has

---

[3] https://github.com/harryliew/WGAN-QC
[4] https://github.com/milesial/Pytorch-UNet
[5] https://github.com/dipuk0506/SpinalNet
[6] https://github.com/fadel/pytorch_ema

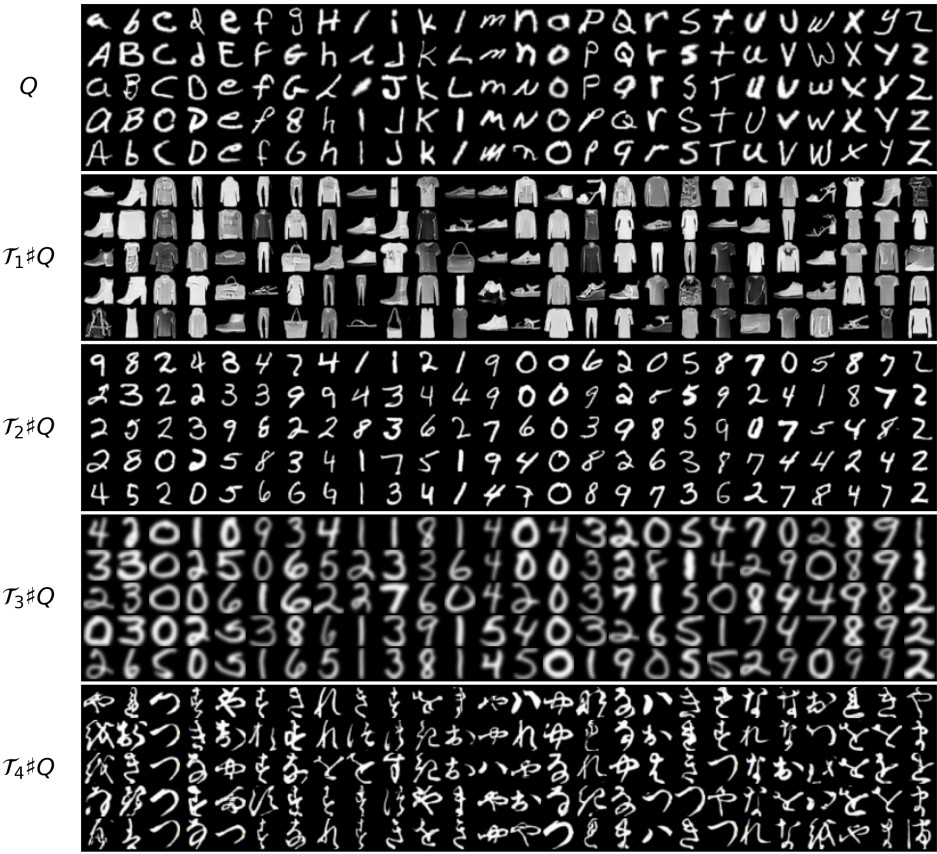

Figure 4: The dataset $Q$ is EMNIST (letters). We show all the datasets pushforwarded towards Fashion-MNIST, MNIST, USPS, KMNIST by OTDD map. The OTDD map is solved by neural OT method.

the same conditional distribution $p(x|y)$ as source label. For example, the sample from top left corner cluster is still mapped to the top left corner cluster, and the label is changed from blue to orange. This map achieves zero transport cost. Since the transport cost is always non-negative, this map is the optimal OTDD map. However, Asadulaev et al. (2022); Bunne et al. (2022a) enforce mapping to preserve the labels, so with their methods, the blue cluster would still map to the blue cluster. Thus their feature map is highly non-convex and more difficult to learn. We refer to Figure 5 in Asadulaev et al. (2022) for their performance on the same example. Compared with them, our pushforward dataset aligns with the target dataset better.

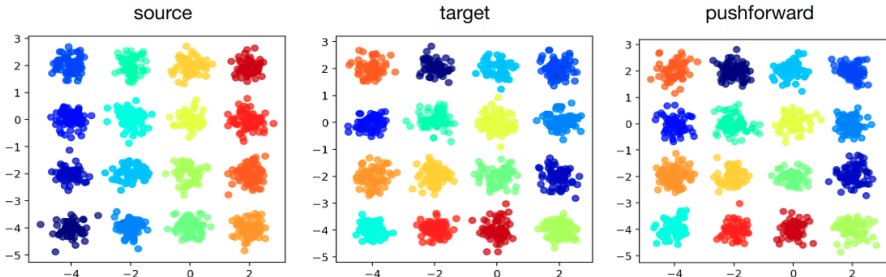

Figure 5: OTDD neural map for 2D Gaussian mixture distributions.

### E.2  McCann's interpolation between datasets

Our OTDD map can be extended to generate McCann's interpolation between datasets. We propose an anolog of McCann's interpolation (1) in the dataset space. We define McCann's interpolation between datasets $P_0$ and $P_1$ as

$$P_t^M := ((1-t)\mathrm{Id} + t\mathcal{T}^*)\sharp P_0, \quad t \in [0,1],$$

where $\mathcal{T}^*$ is the optimal OTDD map from $P_0$ to $P_1$ and $t$ is the interpolation parameter. The superscript $M$ of $P_t^M$ means McCann. We use the same convex combination method in Sec. 3.2 to obtain samples from $P_t^M$. Assume $(x_0, y_0) \sim P_0$, $(x_1, y_1) = \mathcal{T}^*(x_0, y_0)$ and $P_0, P_1$ contain 7, 3 classes respectively, i.e. $y_0 \in \mathbb{R}^7, y_1 \in \mathbb{R}^3$. Then the combination of features is $x_t = (1-t)x_0 + tx_1$, and the combination of labels is

$$y_t = (1-t)\begin{bmatrix} y_0 \\ \mathbf{0}_3 \end{bmatrix} + t\begin{bmatrix} \mathbf{0}_7 \\ y_1 \end{bmatrix}.$$

Thus $(x_t, y_t)$ is a sample from $((1-t)\mathrm{Id} + t\mathcal{T}^*)\sharp P_0$. We visualize McCann's interpolation between two Gaussian mixture distributions in Figure 6. This method can map the labelled data from a dataset to another, and do the interpolation between them. Thus we can use it to map abundant data from an external dataset, to a scarce dataset for data augmentation. For example, in Figure 7, the target dataset only has 30 samples, but the source dataset has 60000 samples. We learn the OTDD neural map between them and solve their interpolation. We find that $P_1^M$ creates new data out of the domain of the original target distribution, which Mixup (Zhang et al., 2018) can not achieve. Thus, the data from $P_t^M$ for $t$ close to 1.0 can enrich the target dataset, and be potentially used in data augmentation for classification tasks.

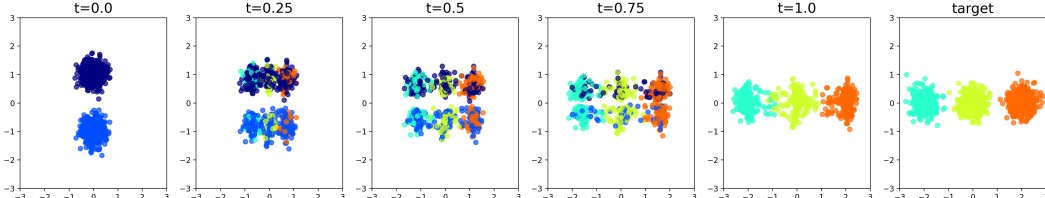

Figure 6: McCann's interpolation for 2D labelled datasets. Each color represents a class. When $t \to 1.0$, the samples within blue classes become less and less, and finally disappear when $t = 1.0$.

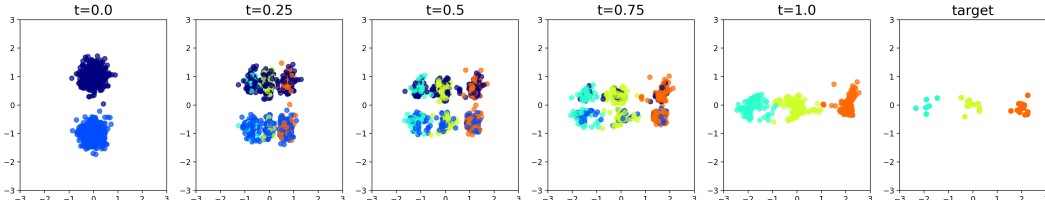

Figure 7: Data augmentation by mapping an external dataset to a few-shot dataset.

