# OpenReview forum: "Generating Synthetic Datasets by Interpolating along Generalized Geodesics"
_NeurIPS.cc/2022/Workshop/SyntheticData4ML — Neurips 2022 SyntheticData4ML_

### Official Review · Reviewer_b2c8 · 2022-10-17
**The submission represents an interesting and motivated approach to generate combinations of datasets for training machine learning models.**

**Rating:** 7
**Confidence:** 4

**Review:**

The authors present a methodology based on optimal transport theory to combine multiple datasets together when training a machine learning model. The proposed method is akin to obtain a convex combination of available datasets in the probability space. The paper not only goes through the theoretical toolkit necessary for solving this problem, but also considers the practical aspects of combining datasets together; e.g., how to combine datasets with a different number of labels. The core of the paper is to construct the optimal transport map between different datasets, for which the authors suggest two original methods, which might be of interest in it and of themselves. Results on 5 different real datasets show how both the neural map and the barycentric projection approach for synthesizing multiple datasets achieve a competitive accuracy with respect to existing methods, as well as their connection to the $(2,\nu)$-transport metric.

Overall, I believe this is a solid paper which could be very useful for the community. My minor comments are just about Figure 2 (which needs a more extensive captions to convey the meaning of the ternary plots) and about the theoretical introduction for optimal transport (as someone somewhat familiar but not an expert in the subject I've found a bit dense, but I appreciate 6 pages is not a lot).

---

### Official Review · Reviewer_uYiV · 2022-10-18
**Presents method on generating better training dataset from the set of of available training sets**

**Rating:** 7
**Confidence:** 3

**Review:**

This paper present an interesting approach to generate targeted training set from multiple sets of training data. The idea is simple yet it seems very effective from the presented results. Authors propose to use barycentric projections or optimal transport methods to build a manifold of data space from the available datasets. Further, based on the target set, the best training samples are then sampled from this space. Authors present results mostly on variations of MNIST.

Pros:
1. Choice of problem statement is very relevant.
2. The method seems very effective.

Cons:
1. Since the training set is brought down to be very close to the target data, does it have an adverse affect of the generalization of the final model? This study is very important for the efficacy of the proposed scheme.
2. Experiments on variations of MNIST do not seem sufficient to make a strong claim and more complex and different data subsets should be used.

---

### Official Review · Reviewer_yXnA · 2022-10-18
**Generating synthetic datasets for pre-training using interpolation along generalised geodesics when target domain dataset is known**

**Rating:** 7
**Confidence:** 2

**Review:**

Pros:
* Interesting approach to generate labeled synthetic data from multiple origin datasets, even when the datasets have disjoint labels.
* Good evaluation over several datasets.

Cons:
* The paper is well written but clarity could be improved, especially focusing on non-expert readers.

Minor typos:
* Line 25: datasets by might -> datasets might be
* Line 33: only of those -> only on those
* Line 132: The barycentric projection is be divided

---

### Meta-Review · Area_Chair_XBMX · 2022-10-19

**Recommendation:** Accept